# Clinical Outcomes and Quality of Life after Patent Foramen Ovale (PFO) Closure in Patients with Stroke/Transient Ischemic Attack of Undetermined Cause and Other PFO-Associated Clinical Conditions: A Single-Center Experience

**DOI:** 10.3390/jcm12185788

**Published:** 2023-09-05

**Authors:** Salvatore Evola, Emmanuele Antonio Camarda, Oreste Fabio Triolo, Daniele Adorno, Alessandro D’Agostino, Giuseppina Novo, Eustaquio Maria Onorato

**Affiliations:** 1Catheterization Laboratory, Department of Medicine and Cardiology, Azienda Ospedaliera Universitaria Policlinico “P. Giaccone”, Via del Vespro 129, 90127 Palermo, Italy; cardioevola@gmail.com (S.E.); camardamed93@gmail.com (E.A.C.); orestefabio.triolo@gmail.com (O.F.T.); daniele.adorno91@gmail.com (D.A.); sandrus.dag@gmail.com (A.D.); 2Department Promise, Università di Palermo, UOC Cardiologia, Azienda Ospedaliera Universitaria Policlinico “P. Giaccone”, Via del Vespro 129, 90127 Palermo, Italy; giuseppina.novo@gmail.com; 3University Cardiology Department, Galeazzi-Sant’Ambrogio Hospital, Scientific Institute for Research, Hospitalization and Healthcare (I.R.C.C.S.), Via Cristina Belgioioso 173, 20157 Milan, Italy

**Keywords:** patent foramen ovale, transcatheter closure, transcranial Doppler, migraine aura, quality of life, cryptogenic stroke, paradoxical embolism

## Abstract

Introduction: The aim of this study was to assess clinical outcomes and quality of life after PFO closure in patients with previous stroke/TIA of undetermined cause and in patients with other complex PFO-associated clinical conditions. Methods: Between July 2009 and December 2019 at our University Cardiology Department, 118 consecutive patients underwent a thorough diagnostic work-up including standardized history taking, clinical evaluation, full neurological examination, screening for thrombophilia, brain magnetic resonance imaging (MRI), ultrasound–Doppler sonography of supra-aortic vessels and 24 h ECG Holter monitoring. Anatomo-morphological evaluation using 2D transthoracic/transesophageal echocardiography (TTE/TEE) color Doppler and functional assessment using contrast TTE (cTTE) in the apical four-chamber view and contrast transcranial Doppler (cTCD) using power M-mode modality were performed to verify the presence, location and amount of right-to-left shunting via PFO or other extracardiac source. Completed questionnaires based on the Quality-of-Life Short Form-36 (QoL SF-36) and Migraine Disability Assessment (MIDAS) were obtained from the patients before PFO closure and after 12 months. Contrast TTE/TEE and cTCD were performed at dismission, 1, 6 and 12 months and yearly thereafter. Brain MRI was performed at 1-year follow-up in 54 patients. Results: Transcatheter PFO closure was performed in 106 selected symptomatic patients (mean age 41.7 ± 10.7 years, range 16–63, 65% women) with the following conditions: ischemic stroke (*n* = 23), transient ischemic attack (*n* = 22), peripheral and coronary embolism (*n* = 2), MRI lesions without cerebrovascular clinical events (*n* = 53), platypnea–orthodeoxia (*n* = 1), decompression sickness (*n* = 1) and refractory migraine without ischemic cerebral lesions (*n* = 4). The implanted devices were Occlutech Figulla Flex I/II PFO (*n* = 99), Occlutech UNI (*n* = 3), Amplatzer PFO (*n* = 3) and CeraFlex PFO occluders (*n* = 1). Procedures were performed under local anesthesia and rotational intracardiac monitoring (Ultra ICE) alone. The devices were correctly implanted in all patients. The mean fluoroscopy time was 15 ± 5 min (range = 10–45 min) and the mean procedural time was 55 ± 20 min (range = 35–90 min). The total occlusion rate at follow-up (mean 50 months, range 3–100) was 98.1%. No recurrent neurological events were observed in the long-term follow-up. Conclusions: The data collected in this study demonstrate that percutaneous PFO closure is a safe and effective procedure, showing long-term prevention of recurrent cerebrovascular events, significant reduction in migraine symptoms and substantial improvement in quality of life.

## 1. Introduction

Most patients with isolated PFO are asymptomatic, and the presence of PFO should not be considered per se a pathological finding [1,2,3]. Nevertheless, the persistence of an interatrial opening in adult life potentially leads to a right-to-left shunt (RLS), which represents the route for paradoxical embolism resulting in embolic stroke or transient ischemic attack (TIA) of undetermined source [4]. In particular, PFO has been associated with stroke/TIA of undetermined cause in young people [5]. According to one study, recurrent events decreased upon antithrombotic treatment from 1.17 per 100 person years (95% CI 0.84–1.78) to 0.29 per 100 person years (95% CI 0.02–0.76) after PFO closure [6]. Additionally, PFO may be implicated in the pathogenesis of several other medical conditions, such as platypnea–orthodeoxia [7], decompression sickness in divers [8], obstructive sleep apnea [9,10], paradoxical coronary embolism [11] and, last but not least, migraine syndromes [12,13]. Quite importantly, migraine with aura is associated with PFO to the same extent as cryptogenic stroke, thus raising questions about the possible relation of causality between the two conditions. PFO may be associated with some abnormalities of the interatrial septum like atrial septal aneurysm (ASA) [14] and Chiari’s network/Eustachian valve [15,16,17], enhancing the chance of RLS and increasing the risk of clinically relevant paradoxical embolism at the same time.

Recent published data showed that transcatheter PFO closure is superior to antiplatelet treatment in the prevention of stroke in selected patients under the age of 60 in terms of reducing the risk of recurrent stroke [18,19,20,21].

We set out to thoroughly examine our ten-year single-center experience, focusing particularly on clinical outcomes and quality of life after PFO closure in patients with previous stroke/TIA of undetermined cause on brain MRI and in patients with other complex PFO-associated clinical conditions. The multidisciplinary framework (“PFO team”) wherein shared decision making and an effective management strategy were discussed proved to be crucial.

## 2. Material and Methods

During the period between July 2009 and December 2019, a total of 960 patients with presumed previous cryptogenic cerebrovascular events and other PFO-associated clinical conditions were referred to our Cardiology Department. Of the overall patient population, the present study included 118 appropriately selected PFO patients with large RLS (shower or curtain patterns) and cerebrovascular events of unknown cause documented and verified through brain MRI according to a decisional flowchart algorithm shared by different specialists of our institution to organize a common approach with the aim of reaching a reasonable therapeutical decision (Figure 1). The decision tree tried to identify which features, in association with PFO, increase the risk of relapses. The critical factors for relapse are the PFO size (>2 mm), the size of the shunt and its impact on brain vessels (as quantified through the bubble test with transcranial Doppler), the coexisting atrial septum aneurysm, venous thrombosis (also considering pelvic veins as a possible source of embolism) and coagulation disorders.

Active involvement of the patient in the decision-making process was achieved and documented in an individualized, open, signed informed consent form.

A thorough diagnostic work-up was performed, including standardized history taking, clinical evaluation, full neurological examination, screening for thrombophilia, brain magnetic resonance imaging (MRI), ultrasound–Doppler sonography of supra-aortic vessels and 24 h ECG Holter monitoring. Anatomo-morphological evaluation through 2D transthoracic/transesophageal echocardiography (TTE/TEE) color Doppler and functional assessment through contrast TTE (cTTE) in the apical four-chamber view and contrast-Transcranial Doppler (cTCD) using power M-mode modality were performed to verify the presence, location and amount of right-to-left shunting via PFO or other extracardiac source. ASA was pre-procedurally diagnosed using TTE or TEE as septum primum excursion of >15 mm towards the right or left atrium. TIA was defined as a transient neurological deficit lasting <24 h with complete resolution of symptoms. Stroke consisted of any new neurological deficit lasting >24 h and confirmed through brain magnetic resonance imaging (MRI). The definition of peripheral embolism was ischemia in any end organ other than the brain caused by reduced flow in a particular artery and objectively documented using Doppler flow or angiographic imaging.

Based on the Venice 1999 Consensus protocol, the bubble count was performed twice, during normal breathing and after Valsalva strain [22]. The contrast medium consisted of 10 mL agitated saline, with the addition of 1 cc patient blood via repeated and forceful injection from one syringe to another through a three-way stopcock. The magnitude of RLS was quantified by counting the number of signals in the middle cerebral artery. RLS was classified as “no shunt”, “small” (<10 signals) or “large” (>10 signals). Among large shunts, the “shower” pattern was defined as shunt with more than 25 signals, and the “curtain” pattern as uncountable signals. All patients who were considered for PFO closure had large right-to-left shunting at rest or after Valsalva maneuver on cTTE and cTCD.

“PFO team” meetings were held regularly at our cardiology department, in which neurology/stroke physicians, neuroradiologists, hematologists and implanting and imaging cardiologists convened to discuss clinical data of each individual patient, including medical records, TTE/TEE, brain MRI, cerebral and vertebral MR angiography, carotid Doppler and screening for thrombophilia (Antithrombin, previously called Antithrombin III, Protein C and S, Lupus Anticoagulant, MTHFR with hyperhomocysteinemia, Factor V Leiden, Factor II, Prothrombin Gene Mutation, anti-β-2-Glycoprotein-1 antibodies, anti-Cardiolipin antibodies, D-Dimer). The stroke etiology and morphological risk were evaluated, and a shared decision was made by consensus using the abovementioned decisional flowchart algorithm; additionally, anatomical and clinical risk factors were also taken into consideration for each individual case (Figure 2).

The main criteria for closure were patients with the following: (1) a first stroke/TIA of undetermined cause with high-risk morphology (ASA) PFO or recurrent cryptogenic stroke and high- or low-risk morphology PFO with or without ASA; (2) aura migraine or refractory migraine with or without ischemic brain lesions; (3) decompression sickness; (4) platypnea–orthodeoxia. Recurrent stroke was defined when more than one ischemic stroke or one stroke associated with multiple ischemic lesions of different ages on brain MRI were reported.

The QoL SF-36 and MIDAS questionnaires were given to the patients in order to measure different aspects of the impact of headache for the sample investigated.

Among the 118 patients, 10 were excluded from the study because they refused to provide informed consent, while another 2 patients preferred to be treated medically.

## 3. Statistical Analysis

Categorical variables are presented as numbers and percentages. Continuous variables are expressed as mean ± standard deviation (SD). The mean values obtained from the analysis of the questionnaires, before and after procedure, were compared using Student’s *t*-test for paired samples, in order to assess the statistical significance of improvements in quality of life. The obtained values are expressed as mean ± 1 SD. *p* values < 0.001 were considered statistically significant in all analyses. All calculations were performed with JASP software, version 0.13.1.0, and with MedCalc statistical software Version 22.009.

## 4. Results

In total, 106 patients were accepted for PFO closure (37 men, mean age 43.5 years; 69 women, mean age 40.9 years). Risk factors, such as smoking, diabetes, hyperlipidemia, hypertension and atrial septal anatomy, are outlined in Table 1. Thrombophilia screening was positive in 68% of the patients (47% have one mutation, 24% have two mutations and 29% have three mutations) (Figure 3A). A comparison between the number of genetic mutations (1, 2, 3) and the increasing number of MRI lesions is shown in Figure 3B. The technique of percutaneous PFO closure has been described in detail [23]. The indications for transcatheter intervention and procedural characteristics are summarized in Table 2 and Table 3. Before closure, all patients were receiving antiplatelet therapy, and intravenous antibiotic prophylaxis was given during the procedure. Unlike other interventional experiences worldwide, all implantation procedures were less invasively performed under mild sedation, and local anesthesia was used to numb the groin area where the catheters are inserted. The imaging guidance was obtained through rotational intracardiac monitoring using a 9F-9MHz rotating ultrasound element catheter (Ultra ICE™, Boston Scientific Corporation, San Jose, CA, USA) introduced via the left femoral vein through a 9-Fr pre-curved polyethylene long venous sheath, as described previously [24,25].

Thrombophilia screening was positive in 68% of the patients (47% have one mutation, 24% have two mutations and 29% have three mutations) (Figure 3A).

A comparison between the number of genetic mutations (one, two, three) and the increasing number of MRI lesions is shown in Figure 3B. The indications for transcatheter intervention and procedural characteristics are summarized in Table 2 and Table 3.

Before the release of the device, the position of the occluder was checked using both fluoroscopy and rotational intracardiac echo. Dual antiplatelet therapy (aspirin 100 mg/day and clopidogrel 75 mg/day) was prescribed for three months, continuing single antiplatelet therapy up to six months. The decision to continue single antiplatelet therapy longer than 6 months was left to physician discretion to improve thromboembolic protection. Two mild/moderate residual RLSs without hemodynamic relevance were found. Moderate/severe residual RLS occurred in two cases (2.1%) without any recurrent cerebrovascular event, and DAPT therapy with aspirin and clopidogrel was continued up to 12 months afterwards. In particular, the vast majority of the patients underwent 1-year clinical and instrumental (contrast-TTE + contrast-TDC, brain MRI) follow-up, repeated thereafter every year. The extension of the therapy with a single antiplatelet drug beyond 5 years has been based on estimation of the balance between patients’ overall risk of stroke for other causes and hemorrhagic risk. Conversely, oral anticoagulants were given to those patients with blood coagulation disorders, chronic atrial fibrillation, recurrent deep venous thrombosis and recurrent pulmonary thromboembolism. Infective endocarditis prophylaxis was also recommended for at least 12 months.

A total of 78 out of 106 patients (73.5%) completed the long-term follow-up (mean 3.03 years per patient, range 3 months–8.7 years). The implanted devices were Occlutech Figulla Flex I/II PFO (*n* = 99), Occlutech UNI (*n* = 3) (Occlutech Holding AG, Feldstrasse 22, 8200 Schaffhausen, Switzerland) Amplatzer PFO (*n* = 3) and CeraFlex PFO occluders (*n* = 1). Among the Figulla Flex II PFO occluders, the vast majority were 23/25 mm (55.4%), followed by 27/30 mm (27.2%), 16/18 mm (8.7%) and 31/35 mm (8.7%) (Appendix A). Occlutech UNI occluders with a size of 28.5 × 28.5 mm were implanted in three patients with septum primum fenestrated aneurysms associated with PFO. An Amplatzer^®^ PFO Occluder device (Abbott, 5050 Nathan Lane North, Plymouth, MN, USA) was implanted in three cases and only one patient received a CeraFlex PFO occlude (Lifetech Scientific (Shenzhen) Co., Ltd., Shenzhen, China).

The selection of the device’s size was based upon PFO anatomical features documented through pre-TTE/TEE color Doppler (Appendix A) and intra-operative morphologic assessment (Ultra ICE™, Boston Scientific, Marlborough, MA, USA). For simple PFOs (short tunnel up to 8 mm, without ASA or prominent eustachian valve, with a thickness of the muscular septum up to 6 mm), a 23/25 mmm device was generally implanted, and this represents 54% of our cases submitted to catheter closure. For PFOs associated with large ASA (up to 25%), 27/30 or 31/35 mm PFO devices were individually considered for adequate closure, with particular attention being paid to avoid reaching the free atrial wall or impinging on surrounding structures. In addition, it is noteworthy to mention that incomplete ASA coverage using a device smaller than the ASA extension may often be sufficient to obtain total abolition of the RLS. Equally sized discs devices (UNI) were preferentially used in cases of additional defects on the fossa ovalis in the form of small or single/multiple defects associated with PFO and ASA, taking care to cross the most central hole in the fossa ovalis with the guide wire and deploying the device in there.

The implantation procedure was successful in all patients (100%). The mean fluoroscopy time and mean procedural time were 15 and 55 min, respectively. Nevertheless, some periprocedural complications occurred (femoral hematoma, arteriovenous fistula), the most important and life-threatening lesion being a retroperitoneal hematoma successfully treated with surgery.

The day after closure, TTE contrast color Doppler was repeated to confirm proper positioning of the device and exclude residual shunt, at which point the patient was discharged. Clinical evaluation, full neurological examination, TTE/TEE contrast color Doppler and cTCD were scheduled at 1, 6 and 12 months postoperatively, and yearly thereafter. Postprocedural transient atrial fibrillation occurred in one patient (successfully cardioverted with medical treatment) and migraine progression occurred in two patients (Table 3). Subjective breathlessness or palpitations were also quite common. Neither occluder device embolization nor infective endocarditis occurred in any of our patients. The total occlusion rate at follow-up (mean 51 months, range 7–104) was 98.1%. No recurrent neurological event was observed during the follow-up. Among the 106 patients submitted to PFO closure, 54 underwent one-year follow-up brain MRI, which was unchanged in the vast majority of patients (46, 85%), whilst only 8 patients showed further scanty supratentorial white matter hyperintensities (WMHs) (Figure 4).

Two mild/moderate and two moderate/severe residual right-to-left shunts without recurrent cerebrovascular events were found.

Among the 62 migraineurs, there was a substantial relief of symptoms with a statistically significant (*p* < 0.001) decline in mean migraine days and MIDAS at 12 months postoperatively compared to the basal score (Figure 5A,B).

In terms of QOL SF-36, 67% of the patients improved, while 27% remained unchanged and only 6% worsened (Figure 6A,B, Table 4).

## 5. Discussion

Stroke represents a significant financial burden for healthcare services. The current population of Italy is 60,248,842, and each year, more than 73,000 strokes occur, causing 44.7 deaths per 100.00 inhabitants [26]. Approximately 80% of strokes are ischemic in origin, caused by thrombotic or embolic occlusion of the cerebral arteries. In about 25% of these strokes, the cause is unknown. The cost of stroke in our country is assumed to be EUR 3.195 million (EUR 53 euros per capita).

Catheter-based PFO closure after presumed paradoxical embolism was first described by Bridges et al. in 1992 with the use of a Bard Clamshell septal umbrella [27]. Since then, remarkable advancements in patient selection, trial design, closure system technology, interventional knowledge and skills have been achieved. The results of four randomized control trials were published in 2017 and 2018 [18,19,20,21], resulting in improvement in the PFO management of patients with stroke/TIA of undetermined cause and in a substantial amendment and update of the guidelines worldwide [28,29].

Concomitantly, several new devices with different technical characteristics more suited for percutaneous PFO closure have been made available, recently including bioabsorbable occluders [30] and systems with minimal implantable material or ‘deviceless’ techniques [31].

Moreover, the use of the RoPE score calculator certainly provided some objective guidance for risk stratification [32]. However, it is noteworthy that multidisciplinary evaluation by stroke neurologists, neuroradiologists, hematologists and structural interventional cardiologists may be more valuable, in order to elucidate which populations and anatomical characteristics are associated with the greatest benefit from closure.

Currently, a consensus among eight European scientific societies on key diagnostic, therapeutic and research issues, from the index event to follow-up, has provided two interdisciplinary position papers [33,34]. The first one proposed a shared approach for a rational PFO management offering appropriate strategies. The second position paper provided the first approach to several PFO-related clinical scenarios beyond left circulation thromboembolism and strongly stressed the need for high-quality evidence on these topics. Based on these position documents, the policy in our institution has been to approach patients suffering from candidate PFO-associated syndromes within a multidisciplinary framework, wherein shared decision making in the indication for procedure and appropriate treatment strategy turned out to be key for selecting the appropriate candidate, minimizing the risks to the patient. Recently, evidence-based guidelines from the Society for Cardiovascular Angiography and Interventions (SCAI) came to the same conclusion, pointing out that the decision to perform PFO closure on any patient for any clinical scenario should be highly individualized and nuanced in the context of a mandatory multi-disciplinary team of primary stakeholders, which, most importantly, should include the patient and a neurologist [35].

Undoubtedly, in the long-term, PFO closure is more effective than medical treatment for secondary prevention of stroke, as has been clearly confirmed by systematic reviews and meta-analyses of randomized trials [36]. Recurrent stroke prevention is also paramount from a health economic perspective. Cost-effectiveness analyses comparing PFO closure and medical therapy for cryptogenic stroke patients have shown that closure procedure leads to significant benefits in terms of in quality-adjusted life years gained and potential cost savings, provided that correct patient selection has been accomplished [37,38,39,40,41,42,43,44,45].

Migraine affects around 12% of the general population and represents a major public health problem and a frequent cause of long-term disability [46].

It is associated with aura in approximately one-third of cases [47]. It should also be noted that PFO is present in 30–50% of those who have aura migraine. The pathogenesis of migraine in patients with RLS via PFO remains unclear. Micro embolic load could act as a trigger of migraine attack, provoking a sudden decrease in oxygen saturation in cerebral circulation, triggering cortical spreading depression and migraine attack as a result [48]. Furthermore, vasoactive chemicals (serotonin) might directly enter the systemic circulation through the RLS instead of being inactivated in the lungs. These vasoactive substances might cause aura migraine attacks due to instability or increased excitability of the central nervous system [13]. It has also been demonstrated that platelet aggregation byproducts coming from the venous circulation and bypassing the lung filter might cross the PFO to reach the brain, triggering aura migraine symptoms [49]. In fact, aura migraine symptoms’ response to P2Y12 platelet inhibition correlated almost perfectly with the therapeutic response to subsequent PFO closure [50].

Three randomized controlled trials (MIST, PRIMA and PREMIUM) have all reported negative results [51,52,53]. Conversely, single-center experiences have demonstrated that PFO closure can effectively prevent migraine attacks [54].

Nevertheless, as demonstrated in our experience and based on the fact that PFO closure is very safe and effective nowadays, we firmly believe that it might be beneficial in aura migraine and notably refractory chronic migraine patients by reducing migraine attacks and migraine days with a substantial improvement in health-related quality of life. Furthermore, we should also consider PFO closure in patients with positive response to platelet P2Y12 receptor inhibitors. In our experience, bubble-migraine-positive patients, those presenting aura migraine symptoms soon after cTCD or cTTE, represent the cohort of individuals that will benefit the most.

The key findings of our study include a very high technical success rate of PFO closure and a low complication rate with no deaths and no recurrent neurological events reported so far. Statistically significant improvements in MIDAS and QoL assessed using the SF-36 Health Survey were observed at 12 months follow-up. Overall, our patients reported significantly higher physical vitality, general health, mental health and social functioning. Studies comparing outcomes for patients who had the PFO procedure with those who did not found that health-related quality of life was better in the PFO closure group, with important benefits also reported within the anxiety/depression domain of the questionnaire [43]. In contrast, the non-closure group had significantly lower scores than the closure group on four of the eight SF-36 subscales (physical functioning, role limitation-physical, vitality and general health).

Our study shows that PFO closure is cost-effective and can be performed safely and efficaciously, and the relatively low rates of recurrent neurological events suggest that the therapeutic benefit of PFO closure obtained in the randomized controlled trials is also likely to be seen in our clinical routine practice.

A more proactive approach reevaluating PFO closure, this once-in-a-lifetime effective procedure, in the setting of multidisciplinary collaboration, shared decision making as a method for qualifying only the causal PFO, and open informed consent would be desirable in the upcoming future.

Our study has certain limitations that must be considered. Firstly, the retrospective analysis of the data introduces a theoretical bias associated with such an investigation. Secondly, this is a single-armed design not reporting comparative data. Thirdly, the small sample size may enhance or diminish the effect of PFO closure and the possibility of recall bias for the patient-reported symptoms.

## 6. Conclusions

The data collected in this study demonstrate that percutaneous PFO closure is a safe and effective procedure both in patients with previous stroke/TIA of undetermined cause and in patients with other complex PFO-associated clinical conditions, showing long-term prevention of recurrent cerebrovascular events and significant reduction in migraine symptoms with a considerable improvement in quality of life.

As importantly recommended by evidence-based guidelines, the standardized multidisciplinary approach and the clinical algorithm described in this paper have been key for the proper assessment of causal PFO as a risk factor for cryptogenic cerebrovascular events and other PFO-associated clinical conditions.

## Figures and Tables

**Figure 1 jcm-12-05788-f001:**
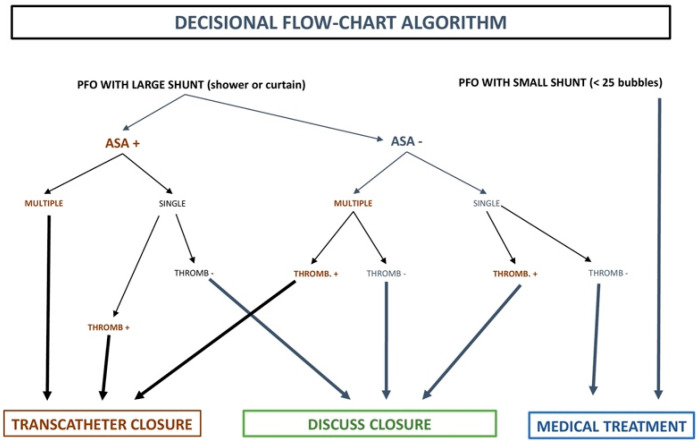
Decisional flowchart algorithm according to the PFO Team. Starting from the size of shunt (large or small), three possible therapeutic conclusions are suggested: on the **right**, indication for medical treatment with antiplatelets underscores the concept that paradoxical embolism in the presence of a small shunt is an unlikely explanation for the stroke and antiplatelets may be sufficiently protective; on the **left**, the recommendation to proceed to transcatheter closure is based on the reasoned assumptions that the patient has had a paradoxical embolic stroke and the risks of the only reasonable medical alternative, namely life-long anticoagulation, outweigh those of the interventional treatment; in the **middle**, the grey area of uncertainty where the decision on which treatment to adopt may depend more on the lifestyle and expectations of the patient than on the true ability of current medical knowledge to provide meaningful advice. ASA, atrial septal aneurysm; Multiple, multiple MRI ischemic lesions; Single, single MRI ischemic lesion; Thromb, thrombophilia.

**Figure 2 jcm-12-05788-f002:**
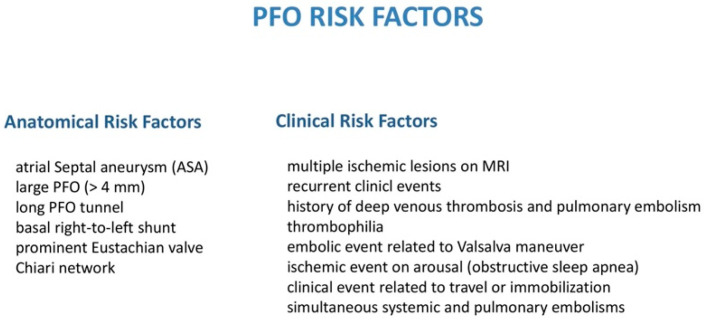
PFO anatomical and clinical risk factors that should be taken into consideration for each individual case.

**Figure 3 jcm-12-05788-f003:**
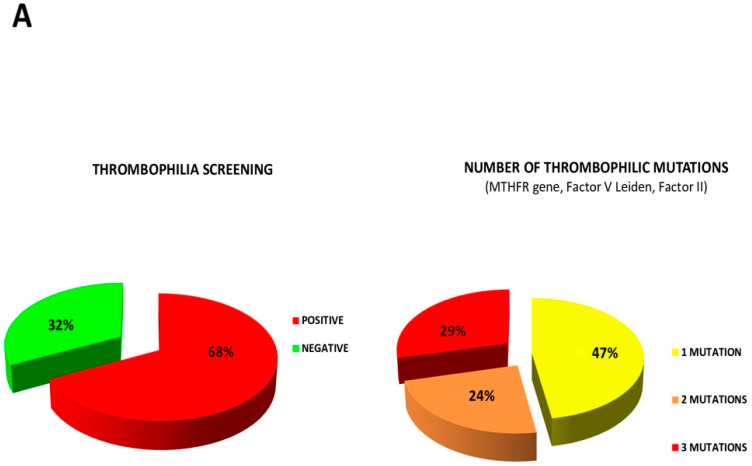
(**A**) Thrombophylic disorders among the total cohort of patients and the number (1, 2, 3) of genetic mutations (%) in the population. (**B**) with an increasing number of genetic mutations, there is a corresponding increasing number of MRI lesions. MRI, magnetic resonance imaging.

**Figure 4 jcm-12-05788-f004:**
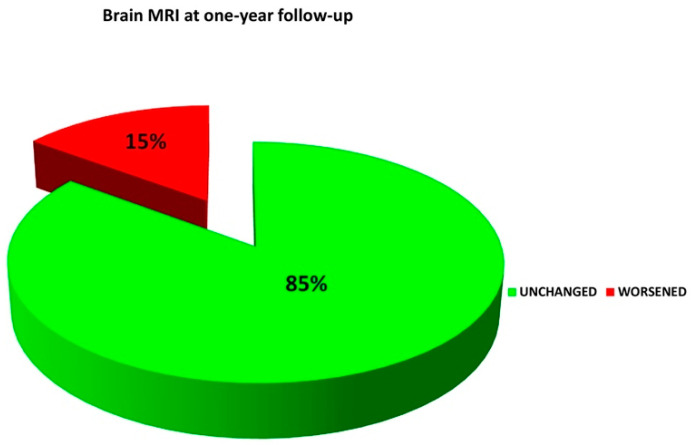
One-year follow-up brain MRI in 54 out of 106 patients.

**Figure 5 jcm-12-05788-f005:**
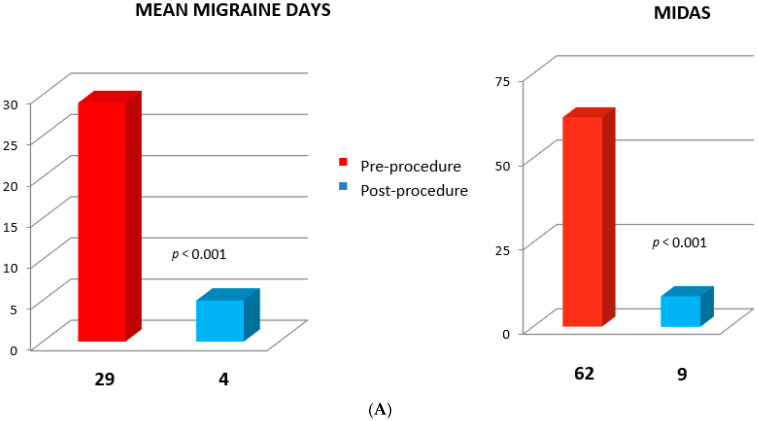
(**A**) Left panel: pre- and post-procedure mean migraine days; right panel: pre- and post-procedure Migraine Disability Assessment Score (MIDAS); (**B**) percentage changes in MIDAS in migraine patients.

**Figure 6 jcm-12-05788-f006:**
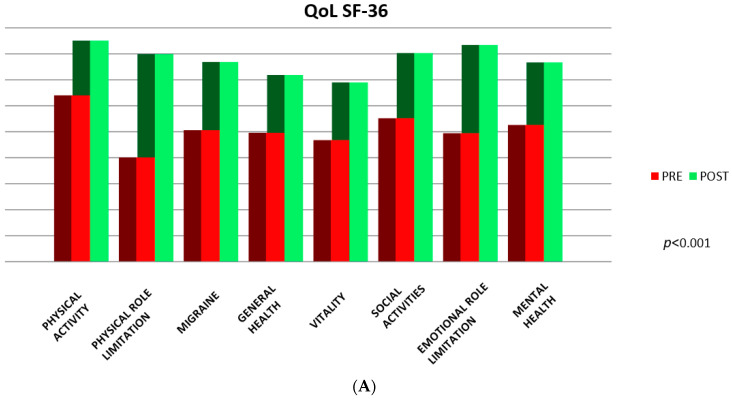
(**A**) Pre- and post-procedure comparison between mean questionnaire values (8 scales) on Quality of Life Short Form-36 (QoL SF-36); (**B**) QoL SF-36 percentage changes (%).

**Table 1 jcm-12-05788-t001:** Demographic characteristics and medical history of the study patients.

Clinical and anatomical characteristics		
**No.**	**106**	
Age, y (range; mean ± SD)	16–62	(41.77 ± 10.78)
Sex, female/male	1.86	
Smoking	72	
Diabetes	19	
Hyperlipidemia	27	
Hypertension	20	
**Atrial septal anatomy, *n* (%)**		
PFO only	20	(18)
PFO and ASA	86	(82)
Fenestrated ASA	6	(6)
Eustachian valve	83	(78)

**Table 2 jcm-12-05788-t002:** Indications for transcatheter PFO closure.

Thromboembolic Events, *n* (%)	100	(94.4)
Stroke	**23**	(21.8)
TIA	22	(20.8)
Brain MRI lesions without TIA/Stroke	53	(50)
Coronary embolism	1	(0.9)
Brachial embolism	1	(0.9)
**“Non-stroke”conditions, *n* (%)**	**6**	**(5.6)**
Decompression sickness	1	(0.9)
Platypnea-orthodeoxia	1	(0.9)
Refractory chronic migraine	4	(3.8)

**Table 3 jcm-12-05788-t003:** Implantation procedure details.

Procedural Success, *n* (%)	106	(100)
**Procedural Characteristics**		
Fluoroscopy time, min (mean)	10–45	(15)
Procedural time, min (mean)	35–90	(55)
**Periprocedural complications, *n* (%)**		
Arteriovenous fistula	12	(11.3)
Venous perforation	1	(0.9)
Retroperitoneal Hematoma	1	(0.9)
Femoral hematoma	5	(4.7)
**Post-procedural complications, *n* (%)**		
Atrial Fibrillation	1	(1)
**Follow-up outcomes, *n* (%)**		
Migraine progression	2	(1.8)

**Table 4 jcm-12-05788-t004:** Comparison between mean and standard deviation values of the 8 scales of the QoL SF-36 questionnaire pre- and 12 months post-procedure.

	Mean Pre-Treatment	Mean Post-Treatment	Standard Dev. Pre-treatment	Standard Dev. Post-treatment	Paired Probability
Physical activity	64.04	85.1	31.32	19.46	*p* < 0.001
Physical role limitation	40.14	79.9	40.65	31.05	*p* < 0.001
Migraine	50.64	76.89	33.32	25.88	*p* < 0.001
General health	49.59	71.89	25.25	22.61	*p* < 0.001
Vitality	46.8	69.02	25.25	22.51	*p* < 0.001
Social activities	55.22	80.31	28.95	20.83	*p* < 0.001
Emotional role limitation	49.44	83.42	42.78	31.01	*p* < 0.001
Mental health	52.61	76.72	26.73	18	*p* < 0.001

## Data Availability

The original contributions presented in this study are included in the article/Appendix A.

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
