# Peer review of "Clinical Outcomes and Quality of Life after Patent Foramen Ovale (PFO) Closure in Patients with Stroke/Transient Ischemic Attack of Undetermined Cause and Other PFO-Associated Clinical Conditions: A Single-Center Experience"

_jcm, 2023, doi:10.3390/jcm12185788_

Round 1

Reviewer 1 Report

Review PFO closure

The authors present their cumulative experience with catheter based PFO closure.

A total of 106 patients was included in the study and follow up.

As evaluated patients have been selected according to a decisional flow chart it would be of interest how many patients have been evaluated in total. Only by this the impact of selection can be estimated.

11% AV fistulas in the groin appear to be a quite high number, at least more than observes after TAVI procedures. Was this do the  larger caliber of the rotational echo used during the procedures?

The follow up has to be described more in detail. Were all 106 patients seen at the 12 months follow up?

A 73% rate of follow up at 51 months is very low and does not allow a quantification of long term complications. Too many patients with complications may hide in the lost to follow up group. For the evaluation of valve prostheses related complications a 95% follow up rate is recommended.

So a 98,1% success rate at 51 months is unlikely when 4 out of 106 patients ( 3,8%) had a remaining shunt.

As with all long term follow up studies the cumulative follow up time ( months or years) and the lost to follow up rate have to be described.

How many patients completed the SF 36 QOL questionnaire and at which time interval?

How can the 15% rate of worsening in the MRI follow up be explained? How does this fit to the conclusion in the abstract that no further neurological events were observed in the long term. Silent  MRI lesions have been an indication  for performing PFO closure.

In the abstract

-

Author Response

Review PFO closure

The authors present their cumulative experience with catheter based PFO closure.

  • A total of 106 patients was included in the study and follow up. As evaluated patients have been selected according to a decisional flow chart it would be of interest how many patients have been evaluated in total. Only by this the impact of selection can be estimated.

Ö Many thanks for your important suggestion. In our ten-year experience working closely with the neurologists, 960 patients with presumed cryptogenic cerebrovascular events and other PFO-associated conditions with right-to-left shunt (RLS) have been evaluated with a multidisciplinary approach and with shared position statements according to the available literature. After a comprehensive and strict evaluation of the available data, we were able to confirm with a high probability in 118 only the causal role of their PFO (significant RLS) that would benefit from percutaneous closure. Of those,10 patients with a clear-cut indication to percutaneous closure refused to be treated. In 2 patients it was impossible in the cath lab to get across the tunnel-like PFO. The remaining 106 patients underwent transcatheter PFO closure successfully.

  • 11% AV fistulas in the groin appear to be a quite high number, at least more than observes after TAVI procedures. Was this do the larger caliber of the rotational echo used during the procedures?

Ö Thanks for your comment. Based on the long-standing experience, the vast majority of femoral vein punctures in our cath lab are accomplished without the use of echo guidance.   

The imaging guidance was obtained by rotational intracardiac monitoring using the rotating ultrasound element catheter (Ultra ICE™, Boston Scientific, USA) introduced via left femoral vein through a 9-Fr pre-curved polyethylene long venous sheath. The right femoral vein was used as the operative line and the introducers’s sizes (7-Fr to 11-Fr) depend on the device chosen for each particular case. So, certainly the need of two femoral veins increases the risk of complications. On the contrary, all implantation procedures were less invasively performed under mild sedation and local anesthesia used to numb the groin area, without the need of general anesthesia and TEE probe.

Among the peri-procedural complications, only one single ominous case of retroperitoneal hematoma (0.9%) successfully treated with surgery occurred; 5 cases of superficial femoral hematomas (4.7%) that resorbed spontaneously were also described. According to the paper published by Snijder RJR et al. [Snijder RJR, Renes LE, Suttorp MJ, Ten Berg JM, Post MC. Percutaneous patent foramen ovale closure using the Occlutech Figulla device: More than 1,300 patient-years of follow up. Catheter Cardiovasc Interv. 2019,93(6),1080-1084], our percentage is certainly inferior in comparison with inguinal hematoma (6.4%) occurred in their series using the same device and sheath technology.

We agree that 12 AV fistulas (11%) appear to be a great number; however, the relatively small sample size may introduce a theoretical bias associated with such a statement. As pointed out before, no ultrasound had been used for catheter insertion. Nevertheless, fistulas resulted from the lack of bony support during manual compression post-introducers removal and often because of low groin puncture. Finally, all fistulas have been ultra-sound guided compressed and spontaneous closure has been achieved.

  • The follow up has to be described more in detail. Were all 106 patients seen at the 12 months follow up?

Ö All 106 patients completed the 12-month follow-up.

A 73% rate of follow up at 51 months is very low and does not allow a quantification of long-term complications. Too many patients with complications may hide in the lost to follow up group. For the evaluation of valve prostheses related complications a 95% follow up rate is recommended.

Ö Conversely to valve-prostheses related complications, the overwhelming majority of patients undergoing PFO closure as secondary stroke prevention are young people, without major co-morbidities. The planned 24-month follow-up program will certainly rule out serious complications like device embolizations or erosions. It’s noteworthy to mention that full device endothelialization occurs in approximately 3 to 4 months. 73% of patients have been followed-up with a mean of 51 months (from 12 to 104 months)

  • So a 98,1% success rate at 51 months is unlikely when 4 out of 106 patients (3,8%) had a remaining shunt. As with all long term follow up studies the cumulative follow up time (months or years) and the lost to follow up rate have to be described

Ö In its most general meaning, complete closure, a procedural success with no measurable residual shunt (RS), differs from effective closure, a successful procedure with none or mild RS (<10 microbubbles at cTCD). Only two among the 4 patients have had moderate RS at 51-month follow-up, whereas the remaining 2 have had a mild insignificant RS and therefore included in the list of procedural success rate (98,1% corresponding to 104 over 106 patients).

In the abovementioned paper of Snijder RJR et al. a moderate/severe RS was found in 5,9% at 1-year follow-up. Furthermore, residual shunt may occur in up to 25% of patients after PFO closure [Deng W et al. Residual Shunt After Patent Foramen Ovale Closure and Long-Term Stroke Recurrence. Ann Intern Med. 2020 Jun 2; 172(11): 717–725] and the association of residual shunt with long-term recurrent stroke/TIA is unknown.

  • How many patients completed the SF 36 QOL questionnaire and at which time interval?

       Ö  105 patients completed the pre- and post- SF 36 QOL questionnaire. Only 1 single patient refused to provide the answers.  

        How can the 15% rate of worsening in the MRI follow up be explained? How does this fit to the conclusion in the abstract that no further neurological events were observed in the long term. Silent MRI lesions have been an indication for performing PFO closure.

      Ö  15% rate of worsening in the MRI follow up can be explained by the finding described by the neuroradiologists as new minimal focal spots corresponding to clinically silent abnormalities visible in deep or periventricular white matter on MRI. Those white matter hyperintensities (WMHs) are particularly apparent on FLAIR MRI. WMHs are interpreted as simple aspecific hyperintensities and not as new ischemic lesions. Conversely, recurrent stroke was defined when more than one ischemic stroke or one stroke associated with multiple ischemic lesions of different ages on brain MRI were reported.  Silent MRI lesions were detected in the professional scuba diver involved in this study.  

"Are all cited references relevant to the research?": In our opinion, the cited references are relevant. If necessary, we would be very grateful for a more precise designation of inappropriate or missing literature.

Reviewer 2 Report

Dear Authors, 

This manuscript has been well written. However, there are some issues that need to be addressed to improve this manuscript.

1.       For introduction part, the definition of PFO is lacking. Can start the first paragraph of Introduction with the definition

2.       Please put available data on PFO i.e. on statistics of PFO in the world/in the country

3.       Please improve the figures & tables: Figure 1, Figure 2 and Table 1, Table 2 and Table 3. Please make sure the font size is standardized, clear and large enough. Please do not screen shoot from the original figures and tables.

4.       Line 165-168: Please state the type of chart showed in the figure 3 A & B. Example: Pie chart showing ….. Same goes to other figures in the MS.

5.       Results: please correct the ‘This In total’

6.       TABLE 2 & 3: spacing is not consistent

7.       Some abbreviations are not stated clearly i.e:  DAPT. Please make sure there are full form plus with abbreviation in bracket. Example: Dual antiplatelet therapy (DAPT).

8.       For table 4, please state what is meant by: 64, 04 and so on. Please rectify in the table caption accordingly.

9.       Line 322-324: please confirm whether the study stated no [36] is a systematic review and meta-analysis of randomised trials? As it seems the article is a Randomized Controlled Trial.

10.   Line 329-330: Please combine this short paragraph with the subsequent paragraph. Please add a brief explanation of the aura migraine

11.   Line 352: Briefly explain the bubble migraine and its relation with PFO

12.   Line 358: spelling of “overall”

13.   In the conclusion paragraph, please add future perspective of this study.

There are some grammatical errors detected. Please correct them accordingly. 

Author Response

Dear Authors, 

This manuscript has been well written. However, there are some issues that need to be addressed to improve this manuscript.

Ö We greatly appreciate your suggestions to ameliorate the manuscript.

1.For introduction part, the definition of PFO is lacking. Can start the first paragraph of Introduction with the definition

Ö Thanks for the suggestion. Appropriate changes (hightlighted in yellow) have been made in the manuscript, accordingly

2. Please put available data on PFO i.e. on statistics of PFO in the world/in the country

Ö Appropriate changes have been made accordingly in the Introduction. PFO Statistics are however available in the first six lines of the Discussion Section.

3. Please improve the figures & tables: Figure 1, Figure 2 and Table 1, Table 2 and Table 3. Please make sure the font size is standardized, clear and large enough. Please do not screen shoot from the original figures and tables.

Ö Appropriate changes have been made, accordingly

4. Line 165-168: Please state the type of chart showed in the figure 3 A & B. Example: Pie chart showing ….. Same goes to other figures in the MS.

Ö Appropriate changes have been made, accordingly

5. Results: please correct the ‘This In total’

Ö Appropriate changes have been made in the manuscript, accordingly: “This” has been deleted

6. TABLE 2 & 3: spacing is not consistent

Ö Appropriate changes have been made, accordingly

7. Some abbreviations are not stated clearly i.e:  DAPT. Please make sure there are full form plus with abbreviation in bracket. Example: Dual antiplatelet therapy (DAPT).

Ö Appropriate changes have been made in the manuscript, accordingly

8. For table 4, please state what is meant by: 64, 04 and so on. Please rectify in the table caption accordingly.

Ö Appropriate changes have been made in the Table and Table caption. 64,04= 64,04%

Line 322-324: please confirm whether the study stated no [36] is a systematic review and meta-analysis of randomised trials? As it seems the article is a Randomized Controlled Trial.

Ö Appropriate changes have been made in the manuscript and in the References, accordingly.The new Reference number is 41 (41. Lai, J.C.L et al.) and it is a systematic review and meta-analysis of randomized controlled trials

Line 329-330: Please combine this short paragraph with the subsequent paragraph. Please add a brief explanation of the aura migraine

Ö Appropriate changes have been made, accordingly. In the modified version of the manuscript, lines 329-330 correspond now to 348-349

Line 352: Briefly explain the bubble migraine and its relation with PFO

Ö Appropriate changes have been made, accordingly. In the modified version of the manuscript, line 352-354 correspond now to 380-386

Line 358: spelling of “overall”

Ö Appropriate changes have been made, accordingly

  1. In the conclusion paragraph, please add future perspective of this study.Ö Appropriate changes have been made, accordingly (lines 421-423).

Reviewer 3 Report

This manuscript addresses a quality of life after PFO closure. The subject matter of this work is laudable and interesting.

I have some comments.

・I think that  SF-36v2 is recently standard  instead of SF-36 for QOL evaluation. Why the author use the previous evaluation criteria?

・Neurological evaluation is also important in this paper. It should indicate when neurological specialists are involved in participation, evaluation, and follow-up.

Author Response

This manuscript addresses a quality of life after PFO closure. The subject matter of this work is laudable and interesting. I have some comments.

・I think that SF-36v2 is recently standard instead of SF-36 for QOL evaluation. Why the author use the previous evaluation criteria?

 Ö The version of the SF-36 questionnaire used by us in this study was the Italian version published in 1998 in the Journal of Clinical Epidemiology [Apolone, G.;Mosconi, P. The Italian SF-36 Health Survey: translation, validation and norming. J Clin Epidemiol.1998;51(11):1025-36]. We chose this version as it is an Italian version that can be freely downloaded from the web, and easily possible to administer to a population of Italian patients in order to be able to obtain greater compliance from them and more reliable answers, overcoming the language barrier of the test. Furthermore, the Italian version of Apolone was translated and published in the following years and the SF-36 v2 was produced. However, comparing the two tests we found the same questions with the same possible answers.

・Neurological evaluation is also important in this paper. It should indicate when neurological specialists are involved in participation, evaluation, and follow-up.

Ö  In our experience, Neurologists are a pivotal part of our PFO closure program for many different reasons: 1) a PFO (better a right-to-left shunt) may be diagnosed with transcranial Doppler (TCD), a neurological tool;  2) PFO is implicated in neurological (cryptogenic stroke, migraine aura, neurological decompression sickness, obstructive sleep apnea) more than cardiological conditions; 3) a multidisciplinary assessment should be performed even when the patient is primarily referred to the cardiologist; 4) in the absence of strong evidence, the decision so as to “close” should only be taken following a consensus of specialists including the neurologist; 5) the neurologist may be involved in monitoring the procedure and in follow-up.